# A Novel Approach toward Less Invasive Multiomics Gut Analyses: a Pilot Study

Adam J. Berlinberg,[a] Ana Brar,[b] Andrew Stahly,[a] Mark E. Gerich,[c] Blair P. Fennimore,[c] Frank I. Scott,[c] ⑩ Kristine A. Kuhn[a]

[a]Division of Rheumatology, Department of Medicine, University of Colorado, Anschutz Medical Campus, Aurora, Colorado, USA
[b]Department of Medicine, University of Colorado, Anschutz Medical Campus, Aurora, Colorado, USA
[c]Division of Gastroenterology and Hepatology, Department of Medicine, University of Colorado, Anschutz Medical Campus, Aurora, Colorado, USA

**ABSTRACT** Newer 'omics approaches, such as metatranscriptomics and metabolomics, allow functional assessments of the interaction(s) between the gut microbiome and the human host. However, in order to generate meaningful data with these approaches, the method of sample collection is critical. Prior studies have relied on expensive and invasive means toward sample acquisition, such as intestinal biopsy, while other studies have relied on easier methods of collection, such as fecal samples that do not necessarily represent those microbes in contact with the host. In this pilot study, we attempt to characterize a novel, minimally invasive method toward sampling the human microbiome using mucosal cytology brush sampling compared to intestinal gut biopsy samples on 5 healthy participants undergoing routine screening colonoscopy. We compared metatranscriptomic analyses between the two collection methods and identified increased taxonomic evenness and beta diversity in the cytology brush samples and similar community transcriptional profiles between the two methods. Metabolomics assessment demonstrated striking differences between the two methods, implying a difference in bacterial-derived versus human-absorbed metabolites. Put together, this study supports the use of microbiome sampling with cytology brushes, but caution must be exercised when performing metabolomics assessment, as this represents differential metabolite production but not absorption by the host.

**IMPORTANCE** In order to generate meaningful metabolomic and microbiome data, the method of sample collection is critical. This study utilizes and compares two methods for intestinal tissue collection for evaluation of metabolites and microbiomes, finding that using a brush to sample the microbiome provides valuable data. However, for metabolomics assessment, biopsy samples may still be required.

**KEYWORDS** intestine, metabolomics, metatranscriptomics, microbiome

The human intestinal microbiome is colonized by trillions of commensal microorganisms (1), and alterations in microbiome composition (or dysbiosis) are associated with a wide variety of inflammatory, metabolic, and infectious human diseases (2–4). The combination of metagenomic, metatranscriptomic, and metabolomic data together with 16S rRNA taxonomic profiling enhances microbial community data, providing functional insights into the role of the human gut microbiome in disease (5). Essential to such multiomics techniques is representative sample collection, where samples can be obtained simply and efficiently and at low cost while providing high-quality data.

The geographic landscape of microbial colonization in the intestine varies based on microanatomic site (6, 7). Thus, studies of the gut microbiome require consideration with regard to the collection method, as a difference between mucosal and luminal inhabitants exists, and different collection methods consequently affect results. A number of various methods have been utilized in the literature, such as fecal sampling,

Address correspondence to Kristine A. Kuhn, kristine.kuhn@cuanshutz.edu.

The authors declare no conflict of interest.

tissue biopsy sample collection, or endoscopic brushing, each with advantages and disadvantages as highlighted in a recent review article (8). The simplest and routinely performed collection method involves fecal sampling, although some argue that it does not represent the true mucosal microbiome (9, 10) sampled from the outer mucosal layer where most microbes inhabit (11, 12). Less frequently will the technique of pinch biopsy sample collection be used due to cost, invasiveness, and subject discomfort. Prior studies have utilized pinch biopsy samples and have identified greater microbial diversity with significant differences in diversity analyses compared to fecal samples (13).

Despite the available collection methods, the fundamental challenge of finding a method of sampling that is noninvasive and cost-effective and provides an accurate depiction of the mucosal microbial environment remains. To address these issues, we propose a method using a cytology brush inserted in the rectum to brush the luminal surface. The collection can be performed in the clinic and requires minimal equipment and training, unlike anoscopy. No sedation or bowel preparation is required, and the procedure can be performed in a few minutes, thus minimizing discomfort for patients. Furthermore, the cost savings are significant compared to endoscopy and aspiration capsule. Described here is a pilot study demonstrating the feasibility of this method compared to endoscopic biopsy sample collection. We find that the cytology brush method provides more bacterial DNA recovery but similar taxonomic information compared to endoscopic biopsy sampling, although metabolomic analysis demonstrates a differential metabolite profile indicative of the diverse mucosal layer, including bacterial- and human-derived metabolites that differ from the deeper biopsy samples.

## RESULTS

**Cytology brush sampling provides improved bacterial DNA recovery and microbial diversity.** The primary goal of this study was to evaluate a novel less-invasive method for microbiome sampling utilizing a cytology brush compared to colon pinch biopsy sample collection. Therefore, we compared the bacterial microbiome in paired cytology brushes to tissue biopsy samples in five individuals undergoing standard-of-care cancer screening colonoscopies. In all, an average of 20.7 million paired-end reads were acquired per sample, with an average of $2.04 \pm 0.47$ Gbp in the biopsy group and $4.22 \pm 3.03$ Gbp in the cytology brush group at a sequence length of 151 bp. The high standard deviation in the cytology brush group reflected a variable amount of tissue collected. Paired-end reads were concatenated, and Kneaddata was used to remove low-quality and human genome-derived reads. After quality control, a total of $14.5 \pm 7.2$ million reads remained in the biopsy samples and $41.7 \pm 31.9$ million reads in the paired cytology brush samples, of which $7.7 \pm 5.1$ million reads remained after removal of human reads in the biopsy group, and $36.3 \pm 29.8$ million reads remained in the cytology brush group (Table S1 in the supplemental material). Of these, a total of $26.5 \pm 11.5\%$ of filtered reads correlated with bacterial sequences in the biopsy samples compared to $59.1 \pm 23.9\%$ of reads in the brushes (Fig. 1A; $P = 0.0256$ by unpaired $t$ test). After removal of human reads, the vast majority of remaining sequences aligned to bacterial reads in both sample types that were used for downstream analysis ($96.1 \pm 2.9\%$ in the biopsy samples and $97.9 \pm 3.5\%$ in the cytology brush samples), with the remaining reads of viral etiology discarded. As predicted, these data support that the brush provides better collection of bacterial DNA than colon biopsy sampling while minimizing host DNA reads, which may be important for studies focused on bacterial taxa identification.

After using the HUMAnN 2.0 pipeline with MetaPhlAn 2.0, alpha diversity was assessed using the MicrobiomeAnalyst software package comparing cytology brush samples to biopsy samples. Alpha diversity on the species level was found to be similar between the two groups, with higher evenness in the cytology brush group. Measures of richness included the observed richness ($P = 0.032$; Fig. 1B) and Chao1 ($P = 0.030$; Fig. 1C), and measures of evenness included the Shannon ($P = 0.017$; Fig. 1D) and

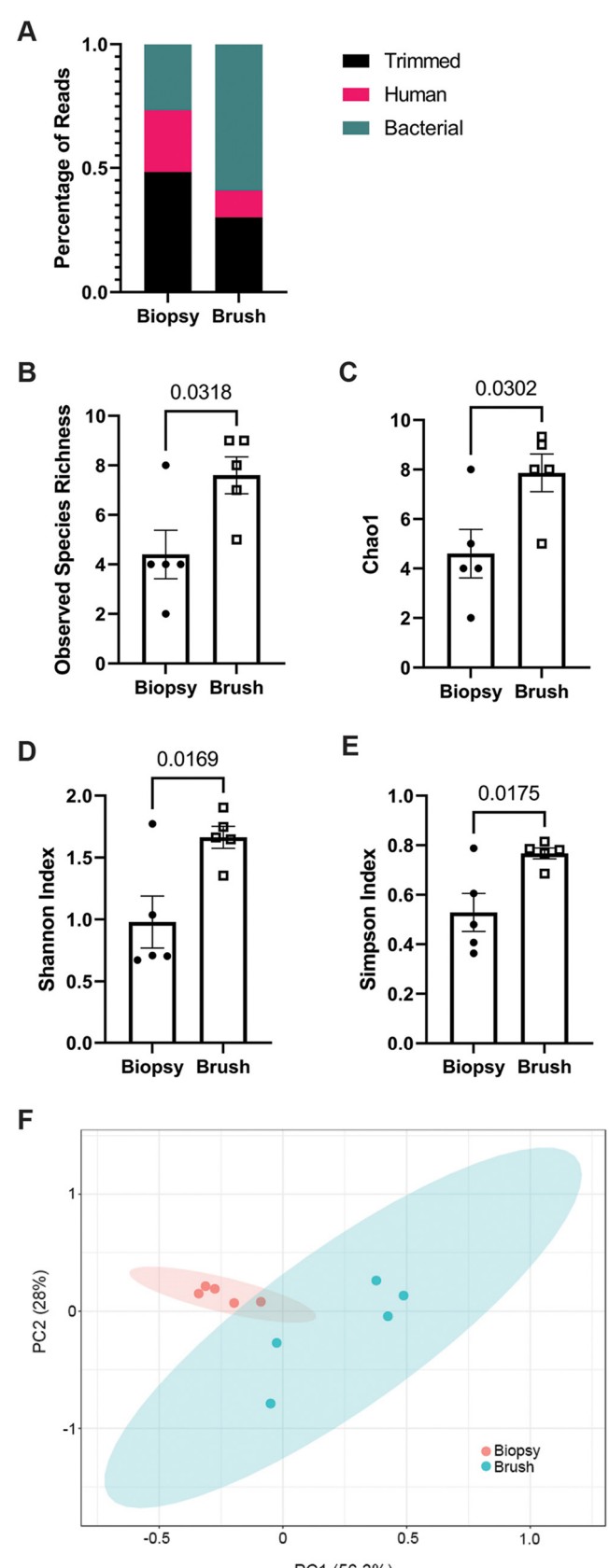

**FIG 1** Cytology brush sampling compared to biopsy samples results in higher abundance of bacteria-derived sequence reads and increased diversity. Metatranscriptomic analysis of bacterial communities

Simpson ($P = 0.017$; Fig. 1E) diversity measures, which were all increased in the cytology brush samples. Beta diversity was assessed as a measure of overall difference between the two samples using the Bray-Curtis index method. An overall difference was determined between the biopsy samples and the cytology brush samples (permutational multivariate analysis of variance [PERMANOVA], $R^2 = 0.366$, $F = 4.62$, $P = 0.028$; Fig. 1F). The top two axes accounted for 84.3% of diversity in the Bray-Curtis index analysis. Based on these findings, we observed an overall trend toward increased alpha diversity in the cytology brush samples compared to biopsy samples as well as a difference in beta diversity between the two collection methods.

**Taxonomic profiles minimally differ between collection methods.** At the class level, the highest prevalence was found to be *Actinobacteria* in the biopsy group (56.1 ± 21.2% versus 22.3 ± 14.9% in the brush group; $P = 0.02$; Table S2), while the highest prevalence in the brush group was *Clostridia* (47.7 ± 10.8% versus 4.1 ± 7.3% in the biopsy group; $P = 0.00007$ and false-discovery rate [FDR] < 0.05). The discordance was persistent at lower levels, such as the top 10 most abundant species (Table S3) and the four statistically significantly different species. The top 10 species were chosen as a simplistic measure of the most abundant species present, eliminating others with very low abundance. The most abundant species in the biopsy group was determined to be *Propionibacterium acnes* (53.3 ± 18.4% versus 13.7 ± 12.2% in the brush group; $P = 0.004$ and FDR < 0.05), while the most abundant species in the brush group was *Faecalibacterium prausnitzii* (17.4 ± 11.6% versus 2.3 ± 3.4% in the biopsy group; $P = 0.023$). Two additional species were found to be statistically significantly different between the two groups: *Streptococcus thermophilus* (higher in the biopsy group; $P = 0.031$) and *Ruminococcus lactaris* (higher in the brush group; $P = 0.042$), although they did not meet our threshold for FDR correction. In all, there was minimal difference in terms of taxonomic profiling between the two collection methods.

**Metatranscriptomics analysis illustrates minimal functional differences between collection methods.** Metatranscriptomics data were analyzed to assess transcribed pathway differences between the two groups. In general, overall Kyoto Encyclopedia of Genes and Genomes (KO) metabolism pathway distribution was largely similar comparing the two groups, with no significant differences identified (Fig. 2A). Within specific individual pathways, the top 10 most abundant hits were then compared (Fig. 2B); two transcripts were significantly different between the two groups: K02703 (*Photosystem II P680 reaction center D1 protein*; $P = 0.02$; higher in the biopsy group) and K02961 (*small subunit ribosomal protein S17*; $P < 0.05$; higher in brush group; Fig. 2C). The top 10 pathways were again analyzed as a measure of overall abundance given the significantly large numbers of pathways identified. Together, these data demonstrate minimal differences in metatranscriptomic profiling between the biopsy sample and cytology brush collection methods.

**Metabolite profiles diverge between sample collection methods.** Given that our microbiome data from cytology brush collection were similar to data from biopsy samples, we next sought to compare collection methods for evaluation of metabolites. In contrast, evaluation of metabolomics data revealed stark differences between the collection methods. Principal-coordinate analysis (PCoA) revealed pronounced separation between the two sample methods with good intragroup correlation, although brushes demonstrated higher variability (Fig. 3A). Assessment of the specific 114 metabolite profiles revealed striking differences between the two collection methods (Fig. S1 and S2), which are further highlighted within the top 25 most abundant metabolites, as

**FIG 1** Legend (Continued)

was performed on paired cytology brush samples and colon pinch biopsy samples from five healthy participants undergoing colonoscopy. (A) Percentage of sequencing reads broken down in terms of bacterial, human, and trimmed reads between sample types. (B to E) Alpha diversity calculated in MicrobiomeAnalyst using the methods of observed species richness (B), Chao1 (C), Shannon (D), and Simpson (E). Values for each subject are shown as a symbol, and bars represent the group means ± standard error of the mean (SEM). Noted $P$ values were determined by unpaired Student's $t$ test. (F) Beta diversity was calculated by the Bray-Curtis dissimilarity index and shown by PCoA.

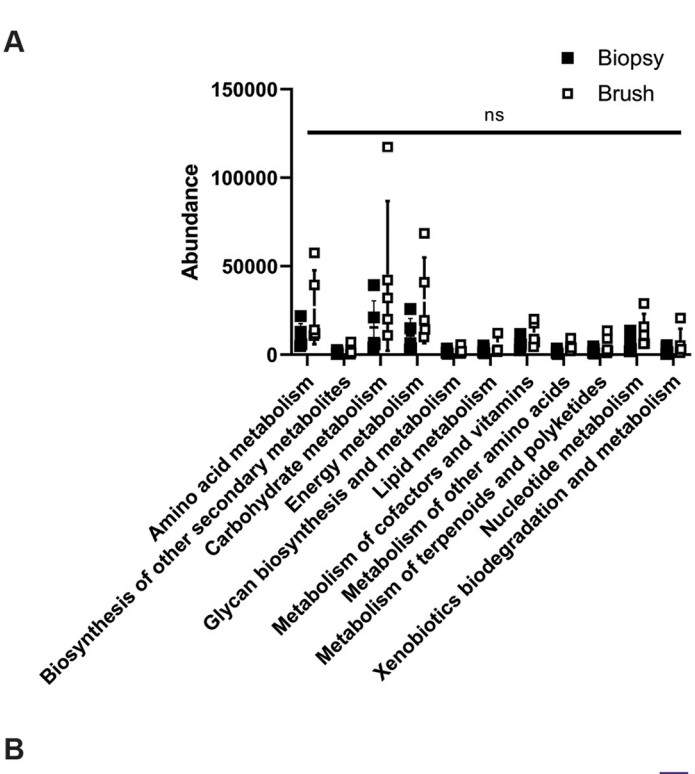

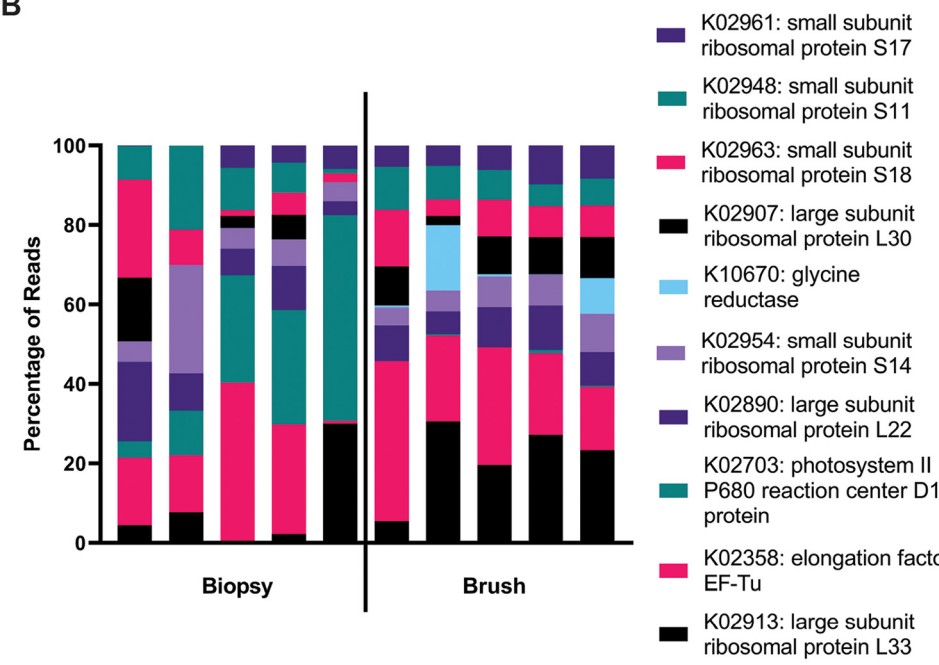

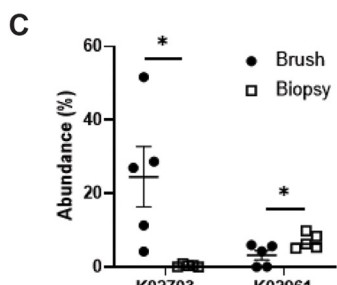

**FIG 2** Metatranscriptomic analyses reveal minimal differences between collection methods. (A) Metatranscriptomics data were assessed for differences in overall pathways between biopsy sample and brush groups, demonstrating

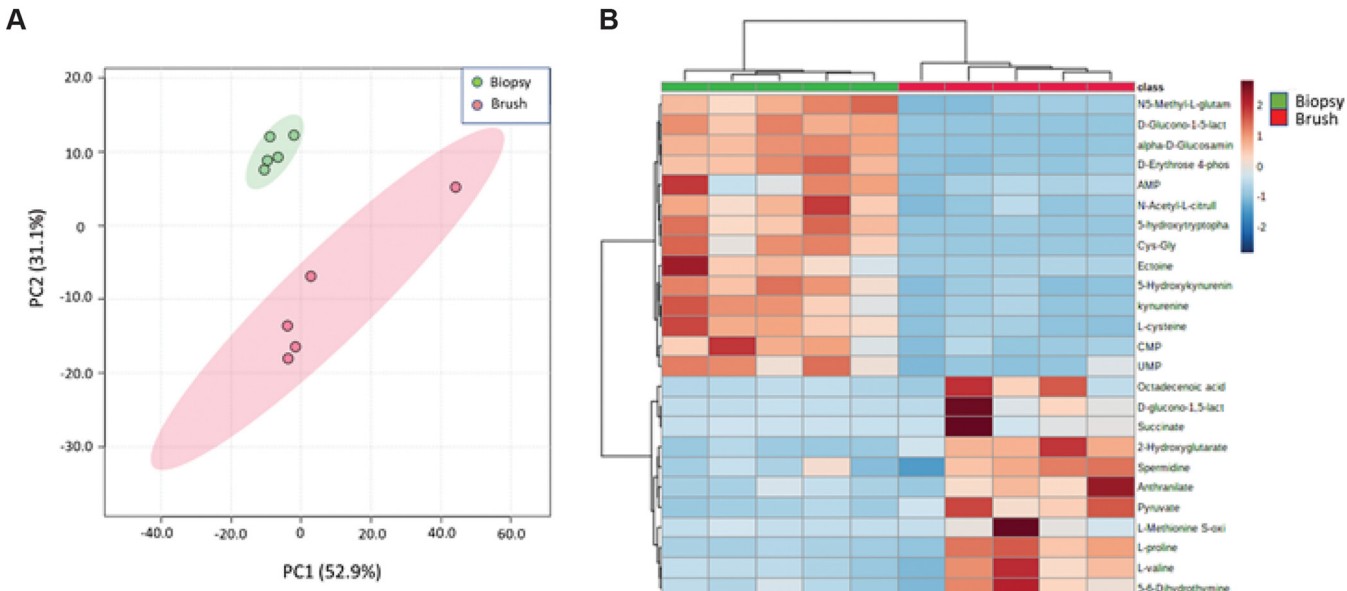

**FIG 3** Assessment of metabolomics reveals striking differences between collection methods. Broad screening of metabolites was performed by UHPLC mass spectrometry comparing biopsy samples and cytology brushes. (A) A PCoA of the metabolite data for all 114 identified metabolites within the two groups is shown. (B) The top 25 identified metabolites based on $t$ test demonstrated as a heat map with a $P$ value of $<0.05$ and an FDR of $<0.05$.

determined by relative tissue abundance (Fig. 3B). After using a fold change of 2.0 and FDR correction of 0.1, a total of 11 metabolites were identified as being significantly different based on a volcano plot (Fig. S2, Table S4). These were largely ubiquitous metabolites in both humans and bacteria, without any bacteria-specific metabolites to compare production versus absorption. Put together, our data reveal differential findings with regard to metabolite profiles between the two sample collection methods of biopsy samples and cytology brush, providing an intriguing difference compared to our metatranscriptomics findings.

## DISCUSSION

Associations between the human gut microbiome and the host immune system are characterized in numerous diseases, including autoimmune disease (14–16). The ability to study this association is predicated on appropriate methods of sample collection and ensuring samples are an accurate representation of the question at hand. In this study, we start to ask the fundamental question of whether sampling techniques result in different answers based on how collection is performed. We aimed to test a new method of less invasive collection that uses cytology brush scrapings of the gut mucosal layer and compared it to colon pinch biopsy sampling by using a multiomics approach. Our findings indicate that a similar metatranscriptomic signature can be identified, but metabolomic profiles indicate very different findings, suggesting that bacterially produced and human-absorbed metabolites are significantly different.

One striking finding is that cytology brush scrapings of the gut mucosal layer demonstrate a significantly increased bacterial read count with transcriptomics sequencing compared to tissue biopsy samples. Our data indicate that 59.1% of the sequencing reads in the brush group correspond to bacterial nucleic acid versus 26.5% of the bi-

**FIG 2** Legend (Continued)

no difference as determined by ANOVA with Kruskal-Wallis *post hoc* test. (B and C) The top 10 most abundant transcriptional pathways (B) were identified and compared between the two groups with a statistical difference in K02703 and K02961 (*, $P < 0.05$), as determined by Kruskal-Wallis *post hoc* test and demonstrated in C comparing the two different sample collection methods of biopsy specimen versus cytology brush.

opsy group (Fig. 1A). This finding suggests that the low bacterial read count from gut biopsy samples with a much higher human nucleic acid count must be removed for assessing the microbiome. Our cytology brush method therefore appears to better sample the mucosal microbiome without human contamination, resulting in a significantly increased bacterial nucleic acid content compared to the biopsy method. A similar result was observed in ileal pouch sampling (17). In general, alpha diversity trends toward an increase in the cytology brush group by all measures, meeting significance just under a $P$ value of <0.05. This is likely a reflection of the small sample size in this pilot study, but these findings also correlate with previous literature assessing microbial diversity across the gut mucus layer compared to the lumen (18).

With regard to beta diversity, our findings suggest greater diversity in the cytology brush group than in the biopsy sample group (Fig. 1F). This finding is likely a reflection of the different layers of the gut being assessed, as the biopsy sample tends to be deeper than the more superficial brush, which may include luminal and mucosal components. Another likely reason for this finding may be the higher sequencing depth as well as bacterial content recovered by the brush, which leads to a higher concentration of bacterially sequenced reads than more human reads that are removed from the biopsy sample group.

Assessing the transcriptomics data demonstrates minor differences with regard to higher-order bacterial identification as well as species identification (Tables S2 and S3 in the supplemental material). The greater microbial diversity on a species level in the brush group and less variability in the biopsy sample group may be due to the more superficial sampling provided by the brush as reflected by the higher alpha and beta diversity shown in Fig. 1. Prior methods have attempted using similar cytology brushes during endoscopic procedures (17, 19, 20), with similar results to ours in that endoscopic brushes resulted in higher bacterial-to-host DNA content with no significant taxonomic differences (17). Other attempts to assess differences in taxonomic profiling among sampling methods have looked at fecal samples compared to colonic biopsy samples using 16S in healthy participants and identified differences in alpha diversity richness (reduced in feces), beta diversity (reduced in feces), and numerous bacterial phyla differences (*Proteobacteria* and *Verrucomicrobia* were significantly reduced in feces) (21). Similarly, prior attempts to compare standard rectal swabs to biopsy samples have identified mixed results having similar taxonomic profiling (22, 23) versus different taxonomic profiling (higher diversity as determined by richness, evenness, and Shannon's diversity and greater amounts of *Lactobacillus* and *Eubacteria* in the swabs than in biopsy samples) (24), although our study appears to be the first to attempt using a cytology brush to scrape the mucosal layer and provide transcriptomic information.

Metatranscriptomics provides functional insights regarding the microbial community, and the end products are the proteins and other biochemicals produced. In this regard, metabolomics profiling can supplement transcriptomic data. Previous studies have identified a number of different microbial-produced metabolites that influence the host immune system, such as short-chain fatty acids, riboflavin metabolites, and tryptophan metabolites (25–27), which highlights the importance of methods to study this interaction. Our findings demonstrate a striking difference between sample methods in the metabolite profile as a whole (Fig. S1) as well as by PCoA and the top 25 most abundant metabolites (Fig. 3). This finding is not unexpected given that different layers of the gut tissue are being analyzed, that is, intestinal tissue with the associated mucous layer in the biopsy samples versus the associated mucous layer itself in the brush samples. The differences in the collections highlight an important caveat in assessing the bacteria-derived metabolome: significant differences exist between metabolites that are produced and what is absorbed, and the results of these data must be considered when critically analyzing future similar studies of gut metabolomics.

**Limitations.** This study has a number of limitations that must be considered. First, as this is a pilot study for proof of concept, the number of participants that were recruited was low, and thus adjustment for potential confounding factors was not performed. These participants were also undergoing routine colonoscopy with bowel prep. Therefore, it is unclear how the microbiome of these participants may be different compared to other studies that rely on stool samples, as we are only assessing the mucosal layer and not all bacteria present in the gut. Repeating this study using participants that have not undergone bowel prep will be of interest. The hypothesis in question relied on metatranscriptomics analysis, which is only one potential 'omics method for microbiome profiling, and others, such as 16S sequencing or metagenomics, could also be performed. Another significant limitation is the lack of technical replicates from the same participant to study how the methods compare with regard to precision. Lastly, there was not a control arm in this study that included a stool sample for additional method comparison, and this was purposely chosen due to the practical inability to perform this analysis prior to bowel prep and concern for temporal differences in sample acquisition.

**Conclusion.** Our results testing use of a cytology brush demonstrate the ability to sample the gut mucosal microbiome. We find that numerous different 'omics approaches are able to be carried out through this sampling method and provide intriguing initial results to be attempted in further studies. Taxonomic profiling demonstrates minor differences, and transcriptional analysis is comparable between the two methods, thus demonstrating the importance of experimental design and hypothesis generation toward sampling method for these types of studies. Metabolite assessment demonstrates a clear difference between the gut mucous layer and the tissue, and this distinction can be used for future studies that aim to understand the role of bacterially derived metabolites and how they are absorbed in the gut. In conclusion, we demonstrate early promising results of this new technique and find new advances in the field of studying gut-microbiome interactions that provide potential for future use.

## MATERIALS AND METHODS

**Sample collection.** Five healthy individuals were recruited through the University of Colorado gastroenterology (GI) clinic between December 2019 and February 2020. Participants were identified from the endoscopy schedule as undergoing routine screening colonoscopy. Inclusion criteria included any subject over the age of 18 undergoing routine colonoscopy. Exclusion criteria included use of antiplatelet drugs or chronic anticoagulation, use of immunomodulatory medications, nonsteroidal anti-inflammatory drugs 7 days before colonoscopy, active malignancy, decompensated cirrhosis, chronic kidney disease on dialysis, or history of inflammatory bowel disease. Patients were identified from the colonoscopy schedule, and written informed consent was obtained just prior to the procedure. This study was conducted according to the principles within the Declaration of Helsinki. All study procedures were approved by the Colorado Multiple Institutional Review Board (protocol number 14-2012).

Patients prepared for colonoscopy by standard protocol the night prior. Following conscious sedation per routine care, one cytology brush (Fisher, 22-281660) was inserted 3 cm beyond the anal verge, pressed against the lateral wall, and rotated two full turns. The brush was then placed into 500 $\mu$L of RNAlater (Thermo Fisher) in a 1.5-mL Eppendorf tube and stored on ice. A second cytology brush was then inserted 3 cm beyond the anal verge against the opposite lateral wall, rotated two times, and placed in 500 $\mu$L of phosphate-buffered saline (PBS) on ice. Lastly, routine colonoscopy was performed with four pinch biopsy specimens obtained at approximately the same depth as the cytology brushes. One biopsy specimen sample was placed in RNAlater, and the remaining three samples were placed in PBS and then on ice. All samples were then taken to the laboratory and frozen at $-80°C$ immediately.

**Microbial RNA isolation, library prep, and metatranscriptomics sequencing.** RNA was isolated using the AllPrep Power fecal DNA/RNA kit (Qiagen). For initial input, brushes in RNAlater were vortexed at maximum speed for 15 s, and 200 $\mu$L of this suspension was used. The manufacturer's protocol was then followed with the exception of incubating samples for 15 min in the presence of lysis buffer and 25 $\mu$L of dithiothreitol (DTT) prior to removal of solid tissue for the biopsy samples, followed by homogenization of samples for both methods in the same manner. Quality control was performed using a Thermo Scientific NanoDrop 2000 spectrophotometer, ensuring 260/280-nm light ratios of >1.8 for all samples. Libraries were then constructed using 5 to 10 ng of RNA for each sample and using the Next Ultra II directional RNA library prep kit with rRNA depletion (New England Bioscience) in a paired-end fashion with 2 × 150-bp paired-end reads. Libraries underwent quality control via tape station prior to multiplexing at a concentration of 4 nM, and sequencing was performed on an Illumina MiSeq platform (San Diego, CA, USA) at the University of Colorado Genomics core with >6 Gbp of data output per sample.

Actual sequencing depth was determined by FastQC and is presented in Table S1 in the supplemental material.

**Data processing and taxonomic analysis.** Manual inspection of sequenced reads was performed using FastQC v0.11.9 for all samples. Paired-end reads were then concatenated, and quality control was conducted with Kneaddata 0.7.5 (http://huttenhower.sph.harvard.edu/kneaddata) using Trimmomatic v0.39 (28) and Bowtie2 v2.3.5 (29) to remove unwanted human genome reads and low-quality sequences. The processed reads were then entered into the HUMAnN 2.0 pipeline (30) using MetaPhlAn v2.0 (31), which does not account for paired-end relationships, with gene profiling abundance performed using the UniRef90 full universal database. Output data in reads per kilobase were then converted to copies per million prior to downstream application using the command humann2_renorm_table. Alpha diversity was determined on a species level using MicrobiomeAnalyst (32, 33) with the observed, Chao, Shannon, and Simpson methods. Beta diversity between the two groups was assessed utilizing Bray-Curtis dissimilarity and visualized with a principal-coordinate analysis (PCoA) plot. PERMANOVA was performed with MicrobiomeAnalyst of the beta diversity clustering. All analyses in MicrobiomeAnalyst were performed on rarefied data.

**Functional analysis.** HUMAnN 2.0 was used with default settings to obtain gene family abundance for each sample individually prior to combining and normalizing based on sequencing depth. Analysis was performed after renaming normalized gene families to Kyoto Encyclopedia of Genes and Genomes (KO) pathways (humann2_regroup_table). Metatranscriptomic abundance was assessed using the functional diversity profile on MicrobiomeAnalyst, and top pathways were identified through read abundance.

**Metabolomics.** Metabolomics analysis by liquid chromatography-mass spectrometry (LC-MS) of tissue collected by cytology brush versus by biopsy was performed by the University of Colorado Metabolomics Core. Cytology brush samples in PBS were spun at 4°C for 10 min at 18,213 relative centrifugal force (rcf). Then, the brush was removed, and centrifugation was repeated. PBS was aspirated and replaced with 100 $\mu$L of ice-cold 5:3:2 methanol:acetonitrile:water (MeOH:MeCN:water [vol/vol/vol]). For biopsy sample tissue, samples were centrifuged twice for 10 min at 18,213 rcf at 4°C, and then PBS was aspirated and replaced with 700 $\mu$L of MeOH:MeCN:water. Extracted samples were vortexed for 30 min at 4°C and centrifuged once as before, and then an aliquot of supernatant was transferred to an autosampler vial for analysis. Samples were analyzed on a Thermo Vanquish ultra-high-performance liquid chromatographer (UHPLC) coupled to a Thermo Q Exactive mass spectrometer. Metabolites were separated on a 5-min $C_{18}$ gradient with positive and negative (separate runs) electrospray ionization. Data acquisition and analysis were performed as previously described (34, 35). Quality control was assessed using technical replicates injected every 10 runs. Resulting .raw files were converted to .mzXML format using RawConverter, and metabolites were assigned and peak areas integrated using Maven (Princeton University) in conjunction with the KO database and an in-house standard library of greater than 600 compounds. The targeted data analysis focused on metabolites involved in central carbon and nitrogen metabolism and yielded measurements of 114 metabolites. No *post hoc* normalization was performed; data are available upon request. Samples were normalized relative to each other based on the same initial starting weight of tissue.

**Data analysis.** Taxonomic and metatranscriptomic profiling was performed using MicrobiomeAnalyst software. For taxonomy alpha diversity, a Student's *t* test was utilized using the methods of Chao, Shannon, and Simpson and the observed species index as they passed Shapiro-Wilk normality tests. Beta diversity was assessed utilizing Bray-Curtis dissimilarity and visualized with a principal coordinate analysis (PCoA) plot. Relative abundances of taxonomic differences were compared using Wilcoxon signed-rank tests and adjusted with an FDR of 0.05. PERMANOVA was performed using MicrobiomeAnalyst software of the beta diversity clustering. Metabolomics assessment was performed using MetaboAnalyst software following log transformation (36). Statistical assessments were done with an analysis of variance (ANOVA) with Kruskal-Wallis *post hoc* or paired *t* test where noted.

**Data availability.** Raw data for the metabolomics and metagenomics sequencing will be made available upon request to the corresponding author. Sequencing data are publicly assessable in the National Library of Medicine's Sequence Read Archive accession PRJNA758430.

## SUPPLEMENTAL MATERIAL

Supplemental material is available online only.

**SUPPLEMENTAL FILE 1**, PDF file, 0.6 MB.

## ACKNOWLEDGMENTS

We acknowledge Francesca Cendali and Julie Reisz of the University of Colorado School Medicine Metabolomics Core for assistance with metabolomics analysis.

We declare no competing financial interests related to this work.

A.J.B. and K.A.K. conceptualized and designed the study. M.E.G., B.P.F., and F.I.S. recruited the participants. M.E.G., B.P.F., and F.I.S. acquired the samples. A.J.B. and A.B. processed and extracted the samples. A.J.B., A.S., and K.A.K. analyzed and interpreted the data. A.J.B., A.B., and K.A.K. drafted the manuscript with input from A.S. All authors contributed to the article and approved the submitted version.

Support for this work includes NIH award T32 AR007534 and the Rheumatology Research Foundation Scientist Development Award (A.J.B.), pilot award funding from the University of Colorado Anschutz Medical Campus Genomics Shared Resource and the Cancer Center Support Grant (P30CA046934) and NIH award AR075033 (K.A.K.).

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
