## [Reviewer comments · Microbiology Spectrum]

Microbiology Spectrum

A Novel Approach Towards Less Invasive Multi ‘Omics Gut Analyses: A Pilot Study

Adam Berlinberg, Ana Brar, Andrew Stahly, Mark Gerich, Blair Fennimore, Frank Scott, and Kristine Kuhn

Corresponding Author(s): Kristine Kuhn, University of Colorado Anschutz Medical Campus

Review Timeline:

Submission Date:	December 5, 2021
Editorial Decision:	January 10, 2022
Revision Received:	March 4, 2022
Editorial Decision:	March 9, 2022
Revision Received:	March 9, 2022
Accepted:	March 10, 2022

Editor: Jennifer Auchtung

Reviewer(s): The reviewers have opted to remain anonymous.

Transaction Report:

DOI: <https://doi.org/10.1128/spectrum.02446-21>

January 10, 2022

Dr. Kristine A. Kuhn
University of Colorado Anschutz Medical Campus
1775 Aurora Ct.
Mail Stop B115
Aurora, CO 80045

Re: Spectrum02446-21 (A Novel Approach Towards Less Invasive Multi 'Omics Gut Analyses: A Pilot Study)

Dear Dr. Kristine A. Kuhn:

Thank you for submitting your manuscript to Microbiology Spectrum. Two reviewers have provided constructive suggestions for improving your manuscript. Please carefully address the comments raised by both reviewers. Please pay special attention to the point raised by reviewer #1 regarding sampling depth and address with 1) description of how sampling depth was determined and 2) initial analysis to determine if sampling depth could explain the majority of differences in sequencing observed. When submitting the revised version of your paper, please provide (1) point-by-point responses to the issues raised by the reviewers as file type "Response to Reviewers," not in your cover letter, and (2) a PDF file that indicates the changes from the original submission (by highlighting or underlining the changes) as file type "Marked Up Manuscript - For Review Only". Please use this link to submit your revised manuscript - we strongly recommend that you submit your paper within the next 60 days or reach out to me. Detailed instructions on submitting your revised paper are below.

Link Not Available

Sincerely,

Jennifer Auchtung

Journals Department
Reviewer comments:

Reviewer #1 (Comments for the Author):

Summary

Berlinberg et al describe their findings from an exploratory analysis of rectal mucosal pinch vs brush biopsy samples. They used a 16S, metatranscriptomic, and metabolomic approach to assess for differences in these sampling methods and consider potential implications for future microbiome studies. The manuscript is generally well-written but occasionally overstates the significance of the findings. I have outlined a number of hopefully constructive comments below, outlined by section.

The major criticism I have is that it isn't obvious how the authors planned sequencing depth. Based on the data from Supplemental table 1, it looks like the brush samples were sequenced with ~3x more reads than the mucosal biopsies, which would account for some or all of the observed taxonomic differences, as well as alpha and beta diversity findings. This point warrants a lot more attention/clarification from the authors, especially since the differences seen in metatranscriptomic/metabolomic analyses were less pronounced.

Comments by section

Importance

- sample size is too small to conclude that brush sampling is superior.

Introduction

- p3, line 19: can the authors cite a paper that host genetic metagenomic contamination is specific to fecal specimens? If not, this statement should be rephrased.
- p4, line 2-4: rectal biopsies collected with anoscopy also do not require sedation, bowel preparation or likely more discomfort than mucosal brushings.

Results

- p8, lines 19-20: the conclusion sentence should be rephrased. The authors' data seems to show that punch biopsies have a higher proportion of host DNA (which would be expected based on a larger fraction of host tissue mass of the collected sample), not necessarily that brush samples are superior for bacterial DNA. This finding may be useful for those who wish to limit their sequencing effort to mucosa-adherent bacterial taxa, which may be what the authors intended to highlight. Please rephrase to reflect this subtle difference, something similar to 'brush samples recovered less human DNA and may be useful for studies focused on the taxa and not host sequence data.'
- p8, lines 23-24: I can't tell if the authors adjusted p-values for multiple testing in these analyses. It looks like one alpha measure that incorporates evenness (Shannon) but not another (Simpson) was significant, which suggests that this may not be a robust / reproducible finding. It also isn't clear how the authors can suggest that one sampling method is better than another when only two methods were performed (i.e. no biological replicates from the same participants or a third sampling method to see how robust the finding is).
- p9, lines 7-8: I don't think the authors have presented enough data to claim that brush samples have higher alpha diversity. They may have had numerically higher alpha diversity measures but they didn't meet significance, which is fine. Looking at Figure 2, it does look like the brush samples recovered more species above the minimal abundance thresholds the authors used for analysis. I think this would be a clearer, easier statistic to understand to just include the median [range] of number species detected in each sample type (which is the simplest measure of richness anyway), reiterate the limited sample size, and not worry about torturing such a small dataset.
- p9, lines 21-23: these p-values do not appear statistically significant.
- p10, line 23: Please remove word 'Unfortunately'

Figures/Tables

- Figure 1, panels B-E: the p-values shown don't seem to match those described in the text on pages 8-9? Please correct these values and ensure all figure and manuscript values are correct.
- Figure 1, panel F: I'm confused by the z-axis in this plot, it looks like only PC1 & PC2 are shown? Is there a reason why the z-axis was also projected? If so, it would be clearer if the authors included an axis label, value range and % variance of PC3. If not, would just plot two axes.
- Figure 2: consider removing p-values from these plots, seems like a greater risk of a type II error with such a small sample size than finding a true difference.
- Supplemental Table 1: table should include the number of samples or additional rows detailing the per-sample means for sequencing depth. This is actually a very important value for their findings, sequencing the brush specimens ~3x more than the biopsy specimens could account for a greater number of detected species, which could account for some or all of their observed differences in alpha and beta diversity.

Reviewer #3 (Comments for the Author):

The manuscript by Berlinger et al describes a pilot study comparing brush vs. colonoscopy sampling, specifically using metatranscriptomics and metabolomics. While studies comparing sampling methods are not new, the comparison of metabolomic and metatranscriptomic data is useful, given the increase in using these methods in the microbiome field. Overall, the methods/approach is well-described and the data presented by the authors supports their conclusions. Specific comments below:

- Metatranscriptomic sequences need to be publicly deposited (stated in the manuscript).
- On page 7, the methods mention the use of OTUs. As this study is metatranscriptomic-based, the authors likely did not use OTU-based analyses, but rather direct taxonomic assignment? Please clarify.
- The authors conclude that differences in the metatranscriptomic results between brush and biopsy samples are minimal. While it certainly looks like the differences are less than those observed for metabolite differences, Figure 2 suggests that there are some clear differences when assessing abundance of different bacterial classes. Perhaps this message should be modified to be more nuanced? Similarly, the authors mention in their discussion that their results are similar to those observed in other studies (using 16S or metagenomic methods). Were these results only similar in the broad sense (diversity or summary metrics), or did their taxonomic results also concur? As in, were a brush or other type of sampling likely to observe increased Clostridia compared to biopsies in these studies as well? Or were completely different results observed? Some additional interpretation would greatly aid the context of these results in comparison to other studies out there.
- The authors demonstrate that metabolomic results between brush and biopsy samples are significantly different, and postulate

that these differences are likely a result of human vs bacterial cells represented in the sample. While I agree that this statement makes sense, some additional data (number of specific metabolites ascribed to human vs bacterial in the 2 sampling types? etc) would strengthen this statement.

Staff Comments:

Preparing Revision Guidelines

Please return the manuscript within 60 days; if you cannot complete the modification within this time period, please contact me. If you do not wish to modify the manuscript and prefer to submit it to another journal, please notify me of your decision immediately so that the manuscript may be formally withdrawn from consideration by Microbiology Spectrum.

Reviewer comments:

Reviewer #1 (Comments for the Author):

Summary

Berlinberg et al describe their findings from an exploratory analysis of rectal mucosal pinch vs brush biopsy samples. They used a 16S, metatranscriptomic, and metabolomic approach to assess for differences in these sampling methods and consider potential implications for future microbiome studies. The manuscript is generally well-written but occasionally overstates the significance of the findings. I have outlined a number of hopefully constructive comments below, outlined by section.

The major criticism I have is that it isn't obvious how the authors planned sequencing depth. Based on the data from Supplemental table 1, it looks like the brush samples were sequenced with ~3x more reads than the mucosal biopsies, which would account for some or all of the observed taxonomic differences, as well as alpha and beta diversity findings. This point warrants a lot more attention/clarification from the authors, especially since the differences seen in metatranscriptomic/metabolomic analyses were less pronounced.

We have added information to the manuscript Supplemental Table 1 that we hope will clarify our sequencing. We intended 6 Gbp sequence output per sample, although FASTQC analysis of the data from each sample demonstrated that we obtained 2.04 ± 0.47 Gbp for the biopsy samples and 4.22 ± 3.03 Gbp for the cytology brush samples. This information is added to the Methods (page 5, lines 22-24) and now included in the revised Supplemental Table 1.

Comments by section

Importance

- sample size is too small to conclude that brush sampling is superior.

We appreciate this criticism by the reviewer and have edited the manuscript throughout to be less conclusive about brush sampling being superior to biopsy.

Introduction

- p3, line 19: can the authors cite a paper that host genetic metagenomic contamination is specific to fecal specimens? If not, this statement should be rephrased.

We have omitted this sentence as host contamination is not unique to fecal specimens.

- p4, line 2-4: rectal biopsies collected with anoscopy also do not require sedation, bowel preparation or likely more discomfort than mucosal brushings.

While anoscopy does not require sedation or bowel preparation, it does require specialized equipment and training by the research staff and is more uncomfortable for a participant than a cytology brush. We have revised page 4, lines 2-3 to add "The collection can be performed in the clinic and requires minimal equipment and training unlike anoscopy. No sedation or bowel preparation is required..."

Results

- p8, lines 19-20: the conclusion sentence should be rephrased. The authors' data seems to show that punch biopsies have a higher proportion of host DNA (which would be expected based on a larger fraction of host tissue mass of the collected sample), not necessarily that brush samples are superior for bacterial DNA. This finding may be useful for those who wish to limit their sequencing effort to mucosa-adherent bacterial taxa, which may be what the authors intended to highlight. Please rephrase to reflect this subtle difference, something similar to 'brush samples recovered less human DNA and may be useful for studies focused on the taxa and not host sequence data.'

Thank you for this suggestion as indeed, we intended to highlight maximizing mucosa-adherent bacteria while limiting host DNA. We have edited the sentence on page 8, lines 20-23 to read "As predicted, these data support that the brush provides better collection of bacterial DNA while minimizing host DNA reads compared to colon biopsies, which may be important for studies focused on bacterial taxa identification."

- p8, lines 23-24: I can't tell if the authors adjusted p-values for multiple testing in these analyses. It looks like one alpha measure that incorporates evenness (Shannon) but not another (Simpson) was significant, which suggests that this may not be a robust / reproducible finding. In these analyses, we did not perform adjustments for multiple t-tests as each comparison stands alone and represents different metrics of community diversity. This is the standard approach for microbiome analysis. In all measures of alpha diversity, we confirmed that $P < 0.05$ by Student's t-test comparing biopsy to brush samples. Although the reported p-values in Figure 1 are correct, those in the manuscript text were not. We have corrected the text.

It also isn't clear how the authors can suggest that one sampling method is better than another when only two methods were performed (i.e. no biological replicates from the same participants or a third sampling method to see how robust the finding is).

The reviewer raises a fair critique of the work. We have edited the manuscript throughout to reflect our findings of how the cytology brush collection compares to biopsy without judgement regarding superiority. Furthermore, in the Limitations section, page 14, lines 10-12, we added:

"Another significant limitation is the lack of technical replicates from the same participant to study how the methods compare with regards to precision."

- p9, lines 7-8: I don't think the authors have presented enough data to claim that brush samples have higher alpha diversity. They may have had numerically higher alpha diversity measures but they didn't meet significance, which is fine. Looking at Figure 2, it does look like the brush samples recovered more species above the minimal abundance thresholds the authors used for analysis. I think this would be a clearer, easier statistic to understand to just include the median [range] of number species detected in each sample type (which is the simplest measure of richness anyway), reiterate the limited sample size, and not worry about torturing such a small dataset.

We apologize for the confusion regarding the alpha diversity in which the p-values in the manuscript were incorrect. They have been corrected to be consistent with those reported in Figure 1, where all were <0.05 . Nevertheless, we agree that the sample size of our study limits us from making firm conclusions. For this reason, we concluded that brush samples *trended* towards increased alpha diversity (page 9, lines 13-14 of the edited version).

As for Figure 2, as this reviewer notes below, there are discrepancies between the p-values in the manuscript and those in the figures. Additionally, we agree that the graphs in the original Figure 2 do not add value to the data presented in Supplemental Tables 2 and 3. Therefore, we have omitted the original Figure 2 from the manuscript per the reviewer's suggestion.

- p9, lines 21-23: these p-values do not appear statistically significant.

Again we identified that the p-values reported in Supplemental Tables 2 and 3 are correct while those included in the manuscript text were not. P-values are based on a Wilcoxon signed-rank test and those with an FDR < 0.05 are noted in the manuscript text and Supplemental Tables. This text now corresponds to the edited manuscript page 10, lines 1-7 and reads: "species in the biopsy group was determined to be *Propionibacterium acnes* ($53.3 \pm 18.4\%$ versus $13.7 \pm 12.2\%$ in the brush group, $p=0.004$ and $FDR < 0.05$), while the most abundant species in the brush group was *Faecalibacterium prausnitzii* ($17.4 \pm 11.6\%$ versus $2.3 \pm 3.4\%$ in the biopsy group, $p=0.02$). Two additional species were found to be statistically significantly different between the two groups: *Streptococcus thermophilus* (higher in biopsy group, $p=0.031$) and *Ruminococcus lactaris* (higher in brush group, $p=0.042$) although they did not meet our threshold for FDR correction. In all, there was minimal difference in terms of taxonomic profiling between the two collection methods."

- p10, line 23: Please remove word 'Unfortunately'

We removed the word as the reviewer suggests.

Figures/Tables

- Figure 1, panels B-E: the p-values shown don't seem to match those described in the text on pages 8-9? Please correct these values and ensure all figure and manuscript values are correct. We apologize for the inconsistencies. We have re-performed all statistical analyses and find the ones in the figure to be correct, updating that which is presented in the manuscript text.

- Figure 1, panel F: I'm confused by the z-axis in this plot, it looks like only PC1 & PC2 are shown? Is there a reason why the z-axis was also projected? If so, it would be clearer if the authors included an axis label, value range and % variance of PC3. If not, would just plot two axes.

We have removed the z-axis and created a 2D plot of PC1 versus PC2.

- Figure 2: consider removing p-values from these plots, seems like a greater risk of a type II error with such a small sample size than finding a true difference.

We agree that the figure is misleading, and per this reviewer's previous recommendation, we

have removed the original Figure 2 mentioned here. Rather, the data remain in the Supplemental Figures 2 and 3 with reported p-values and notation for FDR correction.

- Supplemental Table 1: table should include the number of samples or additional rows detailing the per-sample means for sequencing depth. This is actually a very important value for their findings, sequencing the brush specimens ~3x more than the biopsy specimens could account for a greater number of detected species, which could account for some or all of their observed differences in alpha and beta diversity.

Per the reviewer's request, we have added information on the mean \pm SD for the n=5 samples per group with regards to the sequencing depth to Supplemental Table 1. The amount of Gbps sequenced did not differ statistically between the two groups. We hope this helps clarify the data.

Reviewer #3 (Comments for the Author):

The manuscript by Berlinger et al describes a pilot study comparing brush vs. colonoscopy sampling, specifically using metatranscriptomics and metabolomics. While studies comparing sampling methods are not new, the comparison of metabolomic and metatranscriptomic data is useful, given the increase in using these methods in the microbiome field. Overall, the methods/approach is well-described and the data presented by the authors supports their conclusions. Specific comments below:

- Metatranscriptomic sequences need to be publicly deposited (stated in the manuscript).

They are deposited at the National Library of Medicine's Sequence Read Archive under accession SUB 10104885 and the data were publicly released on January 1, 2022.

- On page 7, the methods mention the use of OTUs. As this study is metatranscriptomic-based, the authors likely did not use OTU-based analyses, but rather direct taxonomic assignment? Please clarify.

Yes, the reviewer is correct that we directly assigned taxonomic identities and not OTUs. We edited the methods accordingly.

- The authors conclude that differences in the metatranscriptomic results between brush and biopsy samples are minimal. While it certainly looks like the differences are less than those observed for metabolite differences, Figure 2 suggests that there are some clear differences when assessing abundance of different bacterial classes. Perhaps this message should be modified to be more nuanced? Similarly, the authors mention in their discussion that their results are similar to those observed in other studies (using 16S or metagenomic methods). Were these results only similar in the broad sense (diversity or summary metrics), or did their taxonomic results also concur? As in, were a brush or other type of sampling likely to observe increased Clostridia compared to biopsies in these studies as well? Or were completely different results observed? Some additional interpretation would greatly aid the context of these results in comparison to other studies out there.

Similar to reviewer #1, this reviewer appropriately criticizes the data presented in Figure 2 as being misleading. While the figure suggests clear differences between taxa abundance in the biopsy versus brush samples, the presentation of data in Supplemental Tables 2 and 3 more clearly display the spread of the data by inclusion of SEM, p-value from the Wilcoxon signed-rank test and those with an FDR < 0.05. Within our taxonomic analyses, only one class and one species reached FDR < 0.05 making the taxonomic comparisons between the biopsies and brush samples very similar. Therefore, in this revision, we are omitting Figure 2 since the presentation is misleading, and the data are better represented in the Supplemental Tables.

With regards to the Discussion, we have edited to add more details about prior studies so that a reader may be able to interpret the comparisons to ours. Starting on page 12, line 19, through page 13, line 10, the paragraph comparing taxonomic results now reads:

“Assessing the transcriptomics data demonstrates minor differences with regards to higher order bacterial identification as well as species identification (Supplemental Tables 2 and 3). The greater microbial diversity on a species level in the brush group and less variability in the biopsy group may be due to the more superficial sampling provided by the brush as reflected by the higher alpha and beta diversity shown in Figure 1. Prior methods have attempted using similar cytology brushes during endoscopic procedures(26, 28, 29), with similar results to ours in that endoscopic brushes resulted in higher bacterial to host DNA content with no significant taxonomic differences(26). Other attempts to assess differences in taxonomic profiling among sampling methods have looked at fecal samples compared to colonic biopsies utilizing 16S in healthy participants, and identified differences in alpha diversity richness (reduced in feces), beta diversity (reduced in feces), and numerous bacterial phyla differences (Proteobacteria and Verrucomicrobia were significantly reduced in feces)(30). Similarly, prior attempts to compare standard rectal swabs to biopsy have identified mixed results having similar taxonomic profiling(31, 32) versus different taxonomic profiling (higher diversity as determined by richness, evenness, and Shannon’s diversity and greater amounts of Lactobacillus and Eubacteria in the swabs versus biopsy)(33), though our study appears to be the first to attempt using a cytology brush to scrape the mucosal layer and provide transcriptomic information.”

- The authors demonstrate that metabolomic results between brush and biopsy samples are significantly different, and postulate that these differences are likely a result of human vs bacterial cells represented in the sample. While I agree that this statement makes sense, some additional data (number of specific metabolites ascribed to human vs bacterial in the 2 sampling types? etc) would strengthen this statement.

The reviewer raises an interesting question as to whether the detected metabolites could be identified as human versus bacterial. We reviewed the top 11 most significantly different metabolites (Supplemental Figure 4) and identified that two were bacteria-specific, one was human-specific, and the rest could be produced by microbes or human.

March 9, 2022

Dr. Kristine A. Kuhn
University of Colorado Anschutz Medical Campus
1775 Aurora Ct.
Mail Stop B115
Aurora, CO 80045

Re: Spectrum02446-21R1 (A Novel Approach Towards Less Invasive Multi 'Omics Gut Analyses: A Pilot Study)

Dear Dr. Kristine A. Kuhn:

Thank you for submitting your manuscript to Microbiology Spectrum. Thank you for addressing the majority of the reviewers' concern. As you will see your paper is very close to acceptance. One point that remains unclear in your Methods is whether alpha diversity analysis was performed on rarefied data. While this appears to be the standard pipeline for MicrobiomeAnalyst (ref 18), please add a sentence in the Methods to confirm whether rarefaction was performed. If rarefaction was not performed, please indicate what methods were used to test whether differences in sequence depth affected differences in observed alpha diversity measures. If there was no normalization for sequence depth prior to determination of alpha diversity measures, this must be performed prior to publication, as differences in sequence depth can contribute differences in alpha diversity measures (e.g., reviewed by Willis (<https://doi.org/10.3389/fmicb.2019.02407>)).

Please modify the manuscript along the lines I have recommended. As these revisions are quite minor, I expect that you should be able to turn in the revised paper in less than 30 days, if not sooner. If your manuscript was reviewed, you will find the reviewers' comments below.

When submitting the revised version of your paper, please provide (1) point-by-point responses to the issues I raised in your cover letter, and (2) a PDF file that indicates the changes from the original submission (by highlighting or underlining the changes) as file type "Marked Up Manuscript - For Review Only". Please use this link to submit your revised manuscript. Detailed instructions on submitting your revised paper are below.

Link Not Available

Sincerely,

Jennifer Auchtung

Reviewer comments:

Preparing Revision Guidelines

- point-by-point responses to the issues I raised in your cover letter
- Upload a compare copy of the manuscript (without figures) as a "Marked-Up Manuscript" file.
- Each figure must be uploaded as a separate file, and any multipanel figures must be assembled into one file.

- Manuscript: A .DOC version of the revised manuscript
- Figures: Editable, high-resolution, individual figure files are required at revision, TIFF or EPS files are preferred

Please return the manuscript within 60 days; if you cannot complete the modification within this time period, please contact me. If you do not wish to modify the manuscript and prefer to submit it to another journal, please notify me of your decision immediately so that the manuscript may be formally withdrawn from consideration by Microbiology Spectrum.

Critique: One point that remains unclear in your Methods is whether alpha diversity analysis was performed on rarefied data. While this appears to be the standard pipeline for MicrobiomeAnalyst (ref 18), please add a sentence in the Methods to confirm whether rarefaction was performed. If rarefaction was not performed, please indicate what methods were used to test whether differences in sequence depth affected differences in observed alpha diversity measures. If there was no normalization for sequence depth prior to determination of alpha diversity measures, this must be performed prior to publication, as differences in sequence depth can contribute differences in alpha diversity measures (e.g., reviewed by Willis (<https://doi.org/10.3389/fmicb.2019.02407>)).”

Response: We did perform rarefaction on the data analyzed in MicrobiomeAnalyst. Per the reviewer’s suggestion, we have added to the Methods (page 6, lines 12-13) “All analyses in MicrobiomeAnalyst were performed on rarefied data.”

March 10, 2022

Dr. Kristine A. Kuhn
University of Colorado Anschutz Medical Campus
1775 Aurora Ct.
Mail Stop B115
Aurora, CO 80045

Re: Spectrum02446-21R2 (A Novel Approach Towards Less Invasive Multi 'Omics Gut Analyses: A Pilot Study)

Dear Dr. Kristine A. Kuhn:

Your manuscript has been accepted, and I am forwarding it to the ASM Journals Department for publication. You will be notified when your proofs are ready to be viewed.

Sincerely,

Jennifer Auchtung
Editor, Microbiology Spectrum
